# Effect of Different Compatibilizers on the Properties of Poly (Lactic Acid)/Poly (Butylene Adipate-Co-Terephthalate) Blends Prepared under Intense Shear Flow Field

**DOI:** 10.3390/ma13092094

**Published:** 2020-05-01

**Authors:** Hezhi He, Guozhen Wang, Ming Chen, Chengtian Xiong, Yi Li, Yi Tong

**Affiliations:** 1National Engineering Research Center of Novel Equipment for Polymer Processing, Guangdong Provincial Key Laboratory of Technique and Equipment for Macromolecular Advanced Manufacturing, South China University of Technology, Guangzhou 510640, China; gjoewang@foxmail.com (G.W.); chenm243@163.com (M.C.); xiongchengtian@163.com (C.X.); 2Key Laboratory of Polymer Processing Engineering, Ministry of Education, South China University of Technology, Guangzhou 510640, China; 3COFCO(jilin) Bio-Chemical Technology Co., Ltd., Changchun 130033, China; lycofco1964@sina.com

**Keywords:** toughening, poly(lactic acid), poly(butylene adipate-co-terephthalate), extrusion, compatibilization, intensive shear flow

## Abstract

In this report, poly(lactic acid) (PLA) and Poly(butylene adipate-co-terephthalate) (PBAT) with three kinds of compatibilizers were melt blended under intensive shear flow. A self-made parallel three-screw extruder was developed to generate such flow during the process. Mechanical properties, chemical reactions among PLA, PBAT and compatibilizers, rheological behavior and morphology were investigated. The mechanical tests showed that the notched impact strength of super-tough composite with 10 wt% EGMA is about 20 times than that of pure PLA. The Fourier transform infrared spectroscopy (FT-IR) results showed that the epoxy functional groups or maleic anhydride functional groups of KT-20, KT-915 and EGMA reacted with the hydroxyl groups of PLA or PBAT macromolecules, resulting in a bridge of PLA and PBAT. About rheological properties, the tan δ—angular frequency curves and the η’’- η’ curves confirmed the chemical reactions mentioned above and indicated better compatibility of η’’- η’ between PLA and PBAT, respectively. Meanwhile, the loss modulus and storage modulus—angular frequency curves demonstrated the discrepancy of different compatibilizer components. In particular, from scanning electron microscopy (SEM) images, it can be seen that the phase size and dispersion uniformity of PBAT adjusted by compatibilizer, corresponding to better compatibility that is described in the η’’- η’ curves. The approach for producing super-tough PLA/PBAT/compatibilizer by intensive shear flow provides a viable direction for further improving PLA performance.

## 1. Introduction

Poly(lactic acid) (PLA), a green polymer material, can be synthesized via polymerization of lactic acid which is fermented by glucose [1]. Renewable agricultural resources such as corn can provide inexhaustible resources for the synthesis of PLA [2], and its products are recyclable and compostable [3]. Moreover, PLA products, with excellent mechanical strength, have good bioabsorbability and biocompatibility with the human body [4], which enables more potential applications. However, the broad application prospect of PLA has been limited due to its poor toughness, especially its low notched impact strength.

Poly(butylene adipate-co-terephthalate) (PBAT), a copolymer derived from 1,4-butanediol, adipic acid and terephthalic acid possesses a tunable balance between biodegradation and desirable physical properties. Therefore, PBAT has many excellent properties such as outstanding ductility, high heat resistance and prominent impact resistance [5]. PLA/PBAT blends have received extensive attention from researchers all around the world [6,7,8,9,10] due to the complementary properties of PLA and PBAT. However, S.W.Ko et al. confirmed that PLA and PBAT are immiscible through scanning electron microscopy (SEM) images [11]. Long Jiang et al. also studied the immiscibility of the blends through DMA and DSC testing [12].

Copolymerization, plasticization and blending are three methods commonly industrially used to modify PLA, and can effectively improve the mechanical properties of composites [13,14,15,16], such as toughness and degradability. Among all the processing methods, blending is the most economical and simple approach by adding compatibilizer for better miscibility. Xue Bin [17,18] prepared super-tough PLA/PBS/EGMA composites and found that EGMA, which contains epoxy functional groups, mostly distributed in the phase interface of two incompatible systems, improving the interfacial bonding between the two phases. Teamsinsungvon [19] prepared PLA/PBAT blends using maleic anhydride grafted PLA (PLA-g-MA) by melt processing and found that tensile properties of PLA/PBAT blend were enhanced, because the interfacial adhesion between PLA and PBAT in the compatibilized blend was improved and even the thermal stability of the blends was also improved with the presence of PLA-g-MA as the compatibilizer. Generally, compatibilizers containing maleic anhydride and epoxy functional groups are effective in improving the microstructure between PLA and incompatible toughening agents, obtaining composites with excellent properties.

The shear flow completes the blending of different materials, and the effective utilization of the shear flow during the blending process generally results in excellent performance [20,21]. Sheng-Yang Zhou [21] introduced strong shear flow into the processing of PLA/PBAT blends. It constructed a coexisting structure of tightly arranged shish-kebabs and PBAT fibers at the interface by strong shear flow, which improved the comprehensive mechanics of the material. However, these mentioned equipment with strong shear flow only have a small yield, while the self-made parallel three-screw extruder (L/D = 40, D = 24.5 mm and resident time = 150–180 s at screw speeds of 180 rpm) shown in Figure 1 can produce strong shear flow and has a large yield. Although the size of the equipment may be a small part of the reason that affects the performance of the blend, type and arrangement of the screw elements in the extruder used in this work and the process conditions of the equipment operation are the key factors that affect the performance of the blend. Based on this equipment with fixed type and arrangement of the screw elements as well as fixed process conditions, the effect of three different compatibilizers for PLA/PBAT was investigated to achieve excellent mechanical properties.

## 2. Materials and Methods

### 2.1. Materials

All information and chemical structures of materials used in this work are display in Table 1 and Figure 2, respectively.

### 2.2. Sample Preparation

The PLA was dried in a blast oven at 85 °C for 10 h. PBAT and three compatibilizers were dried at 80 °C for 10 h. Each group of a final blend is mixed uniformly according to the ratio in Table 2 and then added to the extruder. The temperature forming the feed section to the die of the self-made triple screw extruder was set as 100—125—150—175—190—190—190—190—180 °C. The rotation speed was 120 rpm/min, and the blends were pelletized after extrusion. The pelletized material was pressed into a sheet 1 mm thick at 190 °C and 15 MPa after dried completely, and then cut into wafers with a diameter of 25 mm for rheological testing. Standard samples were obtained for mechanical testing and SEM by applying an injection molding machine (105GE Welltec Machinery Ltd., shanghai, China), wherein the temperature from the feed section to the nozzle was 190—190—190—200 °C.

### 2.3. Scanning Electron Microscopy (SEM)

After the sample was immersed in liquid nitrogen for 30 min, brittle fracture was performed, and then the obtained surface was coated with gold for scanning electron microscopy (S-3700N, Hitachi, Tokyo, Japan) observation at an applied voltage of 5 kV.

### 2.4. Fourier Transform Infrared (FT-IR) Spectroscopy

FT-IR spectroscopy (Bruker vector 3, Bruker Optics, Karlsruhe City, Germany) was used to study the molecular group reactions between PLA and PBAT with presence of compatibilizers. Each sample was scanned by FT-IR spectroscopy with a frequency of 64 scans in a range of 400 to 4000 cm^−1^ after getting a small amount of PLA, PBAT, PLA/PBAT Blend, 10 wt% EGMA/Blend, 10 wt% KT-915/Blend, 10 wt% KT-20/Blend and three compatibilizers samples, and pressing the samples into films with a hot stamper.

### 2.5. Rheological Characterization Measurements

The rheological behavior of the PLA/PBAT blend and the PLA/PBAT/compatibilizer blends were investigated by using a rotational rheometer (MCR 302, Anton Paar Gmbh, Graz, Austria). All tests were implemented at a temperature of 190 °C with a scanning frequency ranging from 0.0628 to 628 rad/s, and the strain amplitude was kept at 1%.

### 2.6. Mechanical Testing

Referring to GB/T 1843-2008, the Izod Impact Tester (Zwick5117, Zwick GmbH, Ulmer, Germany) was used to test the impact strength.

## 3. Results and Discussion

### 3.1. Morphological Analysis

The microstructures of the PLA/PBAT/compatibilizer blends observed by SEM are shown in Figure 3. In Figure 3a, the pure PLA section is very smooth and level, while the PLA/PBAT control group in Figure 3b presents island structures which consist of PBAT in a dispersed phase with a circular cross section and the PLA in a continuous phase. In order to make a subtle discussion, four identifiers are used as follows: yellow circles are used to show significant phase gaps with the PLA matrix and approximate circular PBAT dispersed phase morphology; red circles indicate a morphology of tight interfacial bonding between PBAT and PLA; yellow squares represent an irregularly shaped PBAT dispersed phase morphology; and red squares are assumed to have a very small PBAT dispersed phase morphology in this paper. For details, first, it can be observed from the yellow circles in Figure 3b that PBAT dispersed phases are elliptical-like shaped with not only obvious gaps between PBAT and PLA but also significant holes formed by the debond of PBAT. After using the 2.5 wt% dosage of any one of those three compatibilizers, as Figure 3c–e show, the size of PBAT dispersed phases got smaller and phase gaps became less and narrower. When the dosage of compatibilizer is 10 wt%, as shown in Figure 3d’, the holes decrease conspicuously, and the shape of the dispersed phase even gets into a shape that we cannot define. Secondly, from red circles in Figure 3, it is easy to find the perfectly indistinct gaps as the dosage of compatibilizer is just 2.5 wt%. Furthermore, the area of fairly tight interfacial bonding between PBAT and PLA as described above spread more when the usage of compatibilizer was up to 10 wt%; typically we can see the most area like that in Figure 3e’. Thirdly, the yellow squares show the irregular shapes of PBAT dispersed phases which transformed from the elliptical-like shapes into some irregular shapes as shown in Figure 3 by using compatibilizer. At the same time, it is worthy to note that the area of irregular PBAT dispersed phases became bigger and more collective when the usage of the compatibilizer KT-915 was added up to 10 wt%, which is not reasonable because the changes can be seen, from Figure 3d to Figure 3d’ and from Figure 3e to Figure 3e’, that irregular parts got less and got more dispersive with the increase of compatibilizer. In addition, there are some very tiny PBAT dispersed phases that turned up after introducing compatibilizers as the red squares show, and it is interesting that the size of tiny PBAT dispersed phases in KT-915 components is smaller than those in KT-20 and EGMA components. On the other hand, it seems to be a contradiction that the KT-915 component with 10 wt% compatibilizer in Figure 3c shows some PBAT dispersed phases observably bigger than the KT-915 component with 2.5 wt% compatibilizer in Figure 3c’, while KT-20 components and EGMA components display smaller PBAT dispersed phases.

The statistics, calculated using the software called “Image Pro”, of the particle size and dispersion of the blends are shown in Figure 4; they are based on 5 SEM micrographs with 150–250 counts and use a nonparametric regression method to fit the distribution curve. It can be seen that when compatibilizer was added up to 10 wt%, the peaks of size distribution curves moved to the left compared with 2.5 wt% compatibilizer components’ peaks respectively, and the size of distribution curves of the dispersed phases of PBAT were concentrated in lower size values and narrower range. The reason is that the compatibility between two phases was enhanced by the compatibilizer, which drives PBAT from agglomeration into dispersion.

Additionally, it can be seen that when the dosage of compatibilizer is 2.5 wt%, the small dispersed phases, especially the very tiny PBAT dispersed phases described by red circles in Figure 3 and red arrows in Figure 4b of PBAT in the KT-915 component were remarkably more than the KT-20 and EGMA components and the control group, according to the distribution curves from Figure 4a,d,f. Due to the dispersion trait of PBAT in the KT-915 component, the component might have a better mechanical property. When the dosage of compatibilizer is 10 wt%, even though there were a number of very tiny PBAT dispersed phases as depicted by the left arrow in Figure 4c, dramatically bigger PBAT dispersed phases as depicted by the right arrows in Figure 4d could impair its mechanical property. In stark contrast, despite no tiny small PBAT dispersed phases, small PBAT dispersed phases distribute more widespreadly and uniformly. This dispersion trait of PBAT in the EGMA component tends to facilitate the mechanical property of the blend.

It can be seen from Figure 5 that the characteristic absorption peak of maleic anhydride in the KT-915 curve appeared at 1712 cm^−1^ and 1780 cm^−1^ [22,23,24], indicating that its chemical structure is roughly as shown in the Figure 2. Characteristic absorption peaks of the epoxy group in the EGMA curve and the KT-20 curve can be seen at 910 cm^−1^ [25], indicating that the two compatibilizers contain a large number of epoxy groups and the structures of EGMA and KT-20 according to what is shown in Figure 2. The reaction of the reactive functional group with the hydroxyl group necessarily drives the amount of the reactive functional groups to be less, meaning the characteristic absorption peak curves rise. However, there are a large number of C = O groups corresponding to the characteristic peak at 1750 cm^−1^ which shields the characteristic peaks at 1712 cm^−1^ and 1780 cm^−1^ of maleic anhydride.

The Blend component has the lowest peak (blending vibration peak) height at 1407 cm^−1^ [26], which indicates that its hydroxyl group content is the most, and the same characteristic peak of the KT-915 component is significantly higher than that of the Blend component in Figure 6, demonstrating that the reaction between maleic anhydride and the hydroxyl group [27,28] caused a decline of the hydroxyl group content. Similarly, the introduction of the other two compatibilizers also consumes the hydroxyl group so that the peaks at 1407 cm^−1^ are higher than the peak of the Blend component. It is worth noting that the peak at 1407 cm^−1^ of the KT-20 component is the highest, as shown in Figure 6, but in Figure 7, the KT-20 curve is just slightly lower than the EGMA curve at the characteristic peak (910 cm^−1^) of the epoxy group. It means that the content of KT-20 epoxy groups is relatively small. However, the highest characteristic absorption peak of the KT-20 component in Figure 6 means that most of the hydroxyl groups were consumed. Therefore, this contradiction confirmed the presence of maleic anhydride in KT-20, although the characteristic peak of maleic anhydride is shielded by the peak of C = O at 1750 cm^−1^. In summary, the structure of the three compatibilizers can be proved to correspond to the structure shown in Figure 2 and “bridge macromolecules” are exhibited in red dotted boxes formed by chemical reactions among each compatibilizer and PLA or PBAT, as illustrated in Figure 8 in the melting process, as discussed above.

### 3.2. Dynamic Rheology Properties Analysis

In this study, the rheological behavior of the PLA/PBAT/Compatibilizer blends in response to shear frequency via the dynamic rheological technique illustrates the compatibility [29,30,31] between PLA and PBAT. A number of literature articles indicated that the Cole–Cole diagram can judge the compatibility of a blend [32,33,34,35]: If the curve contains one and only one smooth semi-circular arc, the compatibility of the blend is good; if the curve contains two semicircles or only one semicircle with the right end upturned, it indicates poor compatibility. In Figure 9d, the curve of the PLA/PBAT control group typically has two semicircles, indicating that the compatibility of the two phases is very poor, which is also reflected in the significant phase gap in the SEM image shown in Figure 3b. The Cole–Cole curve of the system is no longer a distinct semi-circular shape when the EGMA dosage is 2.5 wt%; just half of the right semicircle exists and the curvature of the arc is larger than that of the blank group. As the amount of compatibilizer increases, the left semicircle becomes more and more flat and the right semicircle no longer shows a downward bending tendency, but a significant upturn, and the slope of the upturn gets larger, which is equivalent to greater dynamic viscosity and loss viscosity. The shape changes of the curves indicate that the chemical reaction between EGMA and PLA or PBAT leads to “bridge macromolecules” as the tight interfacial bonding between PBAT and PLA shown by Figure 3, which significantly improves the compatibility between PLA and PBAT. In Figure 9e, the curves of KT-20 components with the same compatibilizer content gradient in the Cole–Cole diagram present little changes in dynamic viscosity compared with the curves of EGMA components. It shows a remarkable decline of loss viscosity (η’’(EGMA/Blend)> η’’ (KT-20/Blend)), but the compatibility of the PLA and PBAT is also significantly improved due to the very similar changing trend of KT-20 components curves with EGMA components curves. However, in Figure 9f the shape of curves for KT-915 components does not show the same change pattern as that of the KT-20 and EGMA components. The four curves of the KT-915 components with the same compatibilizer content gradient are significantly different: the 2.5 wt% KT-915/Blend curve position is the highest, while the KT-20 and EGMA components with dosage of 2.5 wt% have the lowest curve position. With the increase of KT-915 dosage, the dynamic viscosity and loss viscosity tend to be getting smaller, which is different from the phenomenon that the dynamic viscosity and loss viscosity of the components using EGMA and KT-20 are getting larger. As shown in the average diameters distribution curve of the PBAT dispersed phase in Figure 4b,c and the yellow squares in Figure 3c’, with the increase of the compatibilizer KT915, the subjects of more large-sized dispersions of the PBAT dispersed phases might have negative effects on the ability for deformation of “bridge macromolecules “ and the ability for force transmission of “bridge macromolecules”, since the “bridge macromolecules” are situated between PLA and PBAT. Therefore, the decrease of dynamic viscosity and loss viscosity may be caused by the increase of PBAT dispersed phase size.

Moreover, it can be seen from Figure 9a–c, that in the low frequency region the tanδ (value of the loss factor) decreases as the amount of compatibilizer increases, continuously. The decrease in the tan δ value of the composite indicates that the elastic response of the composite becomes stronger and the viscoelastic response of the composite melt changes from liquid-like behavior to solid-like behavior [36,37]. The low tan δ value reveals less energy loss and better interface strength [38,39,40], which can be verified by the morphology of tight interfacial bonding that is indicated by the red circles in Figure 3. Obviously, the loss factor-angular frequency curve of the KT-915 component is significantly different from the curves of KT-20 and EGMA components. The KT-20 and EGMA components have lower curves than the Blend component over the entire shear frequency range. But the curves of the 2.5 wt% KT-915/Blend component and the 10 wt% KT-915/Blend component are higher than the curve of the Blend component around the peak of tan δ value; additionally, the 10 wt% KT-915/Blend curve is also higher than the Blend curve in the high frequency region. This indicates that simply increasing the ‘bridge macromolecules’ by using more KT-915 compatibilizer may not further improve compatibility and dynamic rheology properties.

Dynamic rheology properties could confirm chemical reactions among compatibilizers, PLA and PBAT molecules corresponding with the FT-IR analysis. For the EGMA component, the loss modulus curve and the storage modulus curve of each component increase with the increase of the scanning frequency, and the complex modulus value and the loss modulus value of the Blend component are smallest in the low frequency region, but the corresponding values of the Blend component in the high frequency region are the largest. The loss modulus is significantly larger than the storage modulus in the high scanning frequency range from the low scanning frequency, and the storage modulus curves and the loss modulus curves are higher with more use of EGMA. The loss modulus value in the high frequency region is smaller than the storage modulus value, and the loss modulus curve and the storage modulus curve are lower as the EGMA dosage increases. It demonstrates that the viscosity response is dominant and the response gets stronger with increasing amounts of EGMA, while the elastic response is dominant in the high frequency region and the elastic response is weaker with increasing EGMA usage. As analyzed by infrared characterization, the reactive functional groups of the compatibilizer chemically bond with the PLA molecular chain and the PBAT molecular chain to form a macromolecule with a longer molecular chain, such a longer molecular chain having a longer segment. The kinematics of the kinematic unit are weaker, so that the material exhibits a more pronounced viscous response in the low frequency region and a more pronounced elastic response in the high frequency region. The storage modulus and loss modulus of the KT-20 component also have similar features. Different from the features of the above two compatibilizer components, KT-915 components cause similar viscoelastic response as caused by the other two compatibilizers only when the dosage is 2.5 wt%. After the dosage of KT915 was added, both storage moduli–the angular frequency curve of the 10 wt% KT-915 component and the loss modulus–are lower than the curves of the Blend component in Figure 10. At the same time, the viscous response is dominant over the entire scanning frequency range. The improvement of storage modulus and loss modulus of the 2.5 wt% KT-915/Blend component and the consumption of hydroxyl groups in the infrared spectrum indicate that maleic anhydride has reacted with both PLA and PBAT macromolecules to form a similar structure (”bridge macromolecules”) with longer molecules and the branched chain shown in Figure 2. However, the storage modulus and loss modulus decreased when the introduction of KT-915 compatibilizer was more than 2.5 wt%, since the maleic anhydride reaction was incomplete, making the ethylene soft segment of KT-915 molecular chain impair the storage modulus and loss modulus.

From the complex viscosity-angular frequency curve in Figure 10d, as the shear frequency increases, the complex viscosity decreases, showing a significant shear thinning phenomenon of the power law fluids. The behavior of power law fluids [41] is more pronounced with the increase of EGMA dosage according to the increasing of complex viscosity with increased EGMA dosage. It contributes to the result mentioned above that larger molecular chains formed by reactions between the epoxy functional group and PLA or PBAT, following the increased chain entanglement of the molecular chains of the system. The complex viscosity curves of the compatibilizer KT-20 component also show the same trend. It can be seen that the complex viscosity curves of 5 wt% KT-20/Blend and 7.5 wt% KT-20/Blend are higher than the Blend curve in the high frequency region, and different from the curves of the EGMA component, which are lower than the Blend curve in the high frequency region. It is likely that the discrepancy in flexibility between the two compatibilizer molecules of EGMA and KT-20 occurred after uniform dispersion by an intense shear field. As shown in Figure 10f, the complex viscosity curve of the KT-915 component at the dose of 2.5 wt% is higher than the Blend component in the low frequency region. Compared with the 2.5 wt% EGMA/Blend curve and the 2.5 wt% KT-20/Blend curve in Figure 10d,e respectively, the 2.5 wt% KT-915/Blend curve is higher than the Blend curve in a larger shear frequency range. However, using more KT-915 led to the result that KT-915 component complex viscosity curves are below the complex viscosity curve of the Blend component over the entire shear frequency range and the complex viscosity of the composite decreases as the KT-915 content increases. The reason for this phenomenon could also be the poor dispersion and distribution of PBAT dispersion phases mentioned above.

### 3.4. Mechanical Properties Analysis

In this section, as seen in Figure 11, The toughness of the PLA/PBAT blend has been significantly improved after the introduction of compatibilizer. The pure PLA notched impact strength was only 2.0 ±0.1 kJ/m^2^ through the same processing. When the 25 wt% PBAT was added, the notched impact strength of the two-component system was just 3.2 ± 0.5 kJ/m^2^. Furthermore, at the minimum compatibilizer amount (2.5 wt%) the components with compatibilizers EGMA, KT-20 and KT-915 were compared with pure PLA. In these cases, the notched impact strength was up to 5.7 ± 1.1 kJ/m2, (2.9 times), 10.5 ± 1.6 kJ/m2 (5.2 times) and 14.6 ± 2.5 kJ/m^2^ (7.3 times), respectively. The remarkable enhancement of toughness is mainly because of the enhanced interfacial bond since the “bridge macromolecules” formed by the chemical reaction promoting the compatibility of the two phases. As the red square in Figure 3c and the distribution curve of the diameters of PBAT dispersed phases in Figure 4b show, the very small-sized PBAT dispersed phases in 2.5 wt% KT-915/Blend were more than those in 2.5 wt% KT-20/Blend and 2.5 wt% EGMA/Blend. The feature of PBAT dispersed phases in 2.5 wt% KT-915/Blend contributes to the best impact strength among the blends with 2.5 wt% compatibilizer. With the gradual increase of the amount of the compatibilizer, the impact strength of each component was obviously improved. When the amount of compatibilizer is 5 wt% or 7.5 wt%, impact strength of the KT-20 component is better than the KT-915 or the EGMA component, which is probably due to the two kinds of reaction functional groups. When the amount of each compatibilizer (EGMA, KT-20 and KT-915) is the largest (10 wt%) respectively, the notched impact strength is increased to 38.8 ± 1.4 kJ/m^2^ (up 19.4 times), 35.8 ± 2.3 kJ/m^2^ (up 17.9 times) and 29.0 ± 3.6 kJ/m2 (up 14.5 times). As the red square and circle in Figure 3e’ and the distribution curve of the diameters of PBAT dispersed phases in Figure 4g show, the small-sized PBAT dispersed phases in 10 wt% EGMA/Blend are more than those in 10 wt% KT-20/Blend and 10 wt% KT-915/Blend. The feature of PBAT dispersed phases in 10 wt% EGMA/Blend contributes to the dramatic improvement of impact strength.

## 4. Conclusions

In this paper, a series of blends consisting of three different compatibilizers and PLA/PBAT was prepared by a self-made parallel three-screw extruder with intense shearing flow. The impact resistance, compatibility and microscopic morphology of different blends with the same compatibilizer content gradient were studied and compared. The results of FT-IR spectroscopy indicated that the epoxy groups or maleic anhydride groups in the three compatibilizers reacted with the hydroxyl groups of PLA or PBAT, and the changes of storage and loss modulus of dynamic rheology also confirmed the reaction. The occurrence of the micro-interface bond improvement also reflects the effect of the reaction. Comparison of the dynamic rheological properties of the different compatibilizer components manifests that the introduction of the compatibilizer promotes the compatibility of PLA and PBAT, moreover, the SEM cross-section also demonstrates the improvement of compatibility. By analyzing the SEM cross-section morphology, the interface bonding between the PBAT phase and the PLA matrix phase caused by compatibilizer is the main factor for toughness. Furthermore, the size and size uniformity of the PBAT phase play a nonnegligible effect on the toughening. Finally, the smaller size, better uniformity of size of PBAT phase and tighter interface bonding could have synergistic effect on improving toughness, which matches the interesting change in impact strength of different compatibilizer components shown in Figure 11.

## Figures and Tables

**Figure 1 materials-13-02094-f001:**
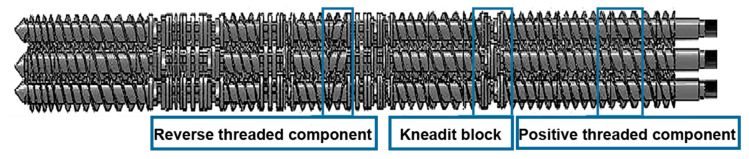
Screw assembly and arrangement of the self-made parallel three-screw extruder.

**Figure 2 materials-13-02094-f002:**
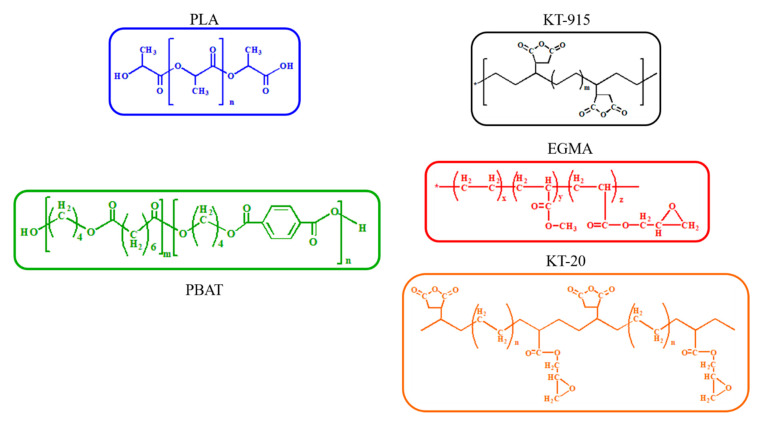
Chemical structure of PLA, PBAT and three compatibilizers.

**Figure 3 materials-13-02094-f003:**
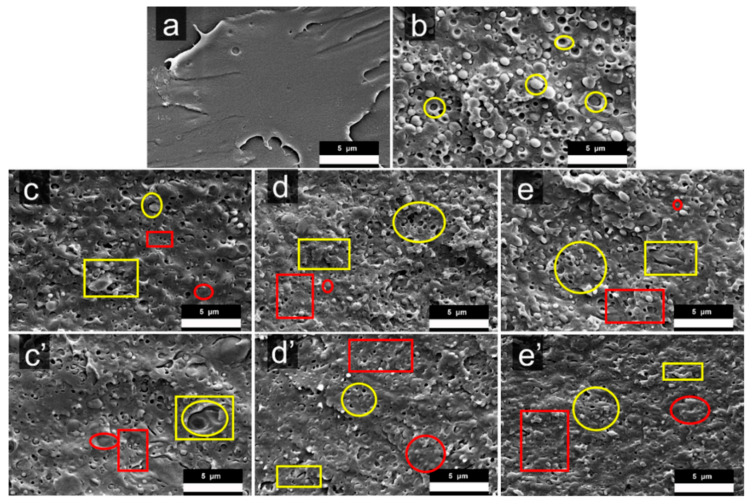
Scanning electron microscopy (SEM) micrographs of cross-sections (**a**) neat PLA, (**b**) Blend, (**c**) 2.5 wt% KT-915/Blend, (**c’**) 10 wt% KT-915/Blend, (**d**) 2.5 wt% KT-20/Blend, (**d‘**) 10 wt% KT-20/Blend, (**e**) 2.5 wt% EGMA/Blend and (**e’**) 10 wt% EGMA/Blend.

**Figure 4 materials-13-02094-f004:**
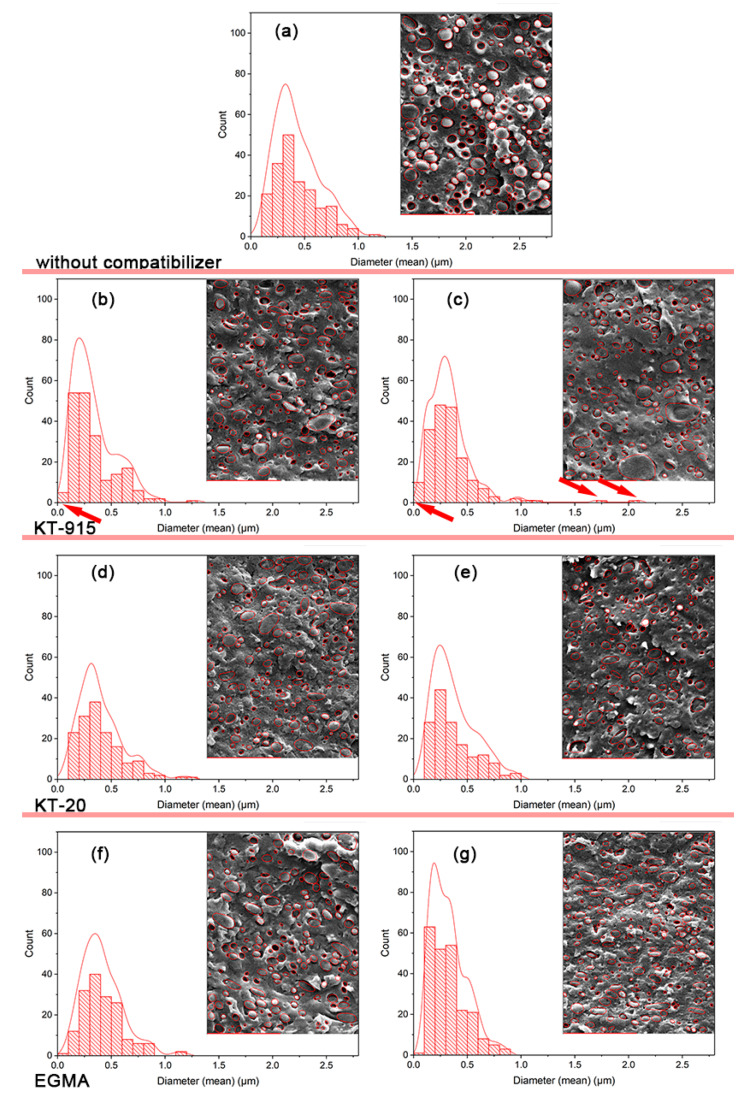
The statistical analyses with respect to the distribution of the diameters of PBAT dispersed phases: (**a**) Blend, (**b**) 2.5 wt% KT-915/Blend, (**c**) 10 wt% KT-915/Blend, (**d**) 2.5 wt% KT-20/Blend, (**e**) 10 wt% KT-20/Blend, (**f**) 2.5 wt% EGMA/Blend and (**g**) 10 wt% EGMA/Blend.

**Figure 5 materials-13-02094-f005:**
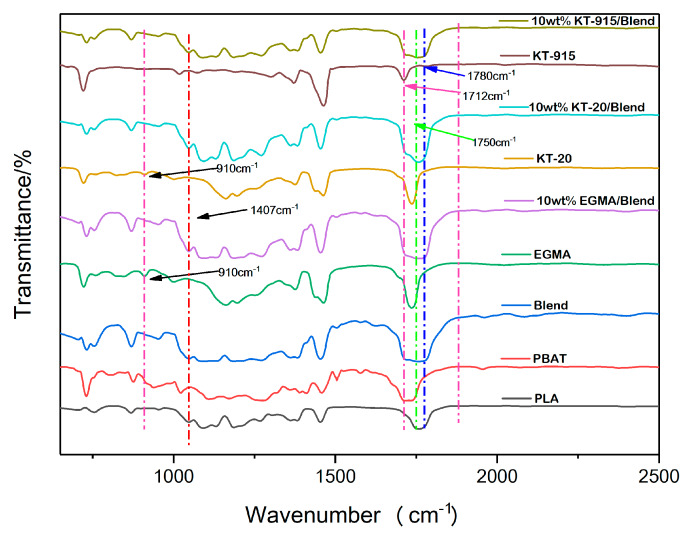
Infrared characteristic peak of each component.

**Figure 6 materials-13-02094-f006:**
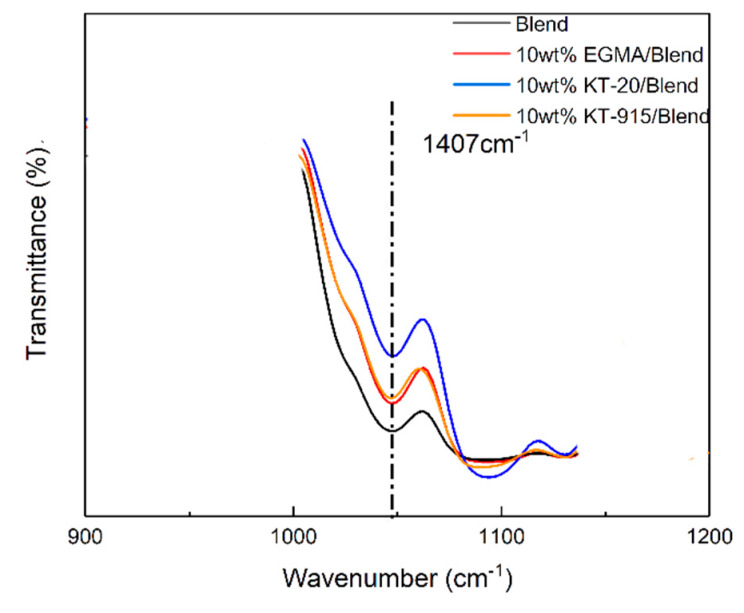
Characteristic absorption peak at 1407 cm^−1^.

**Figure 7 materials-13-02094-f007:**
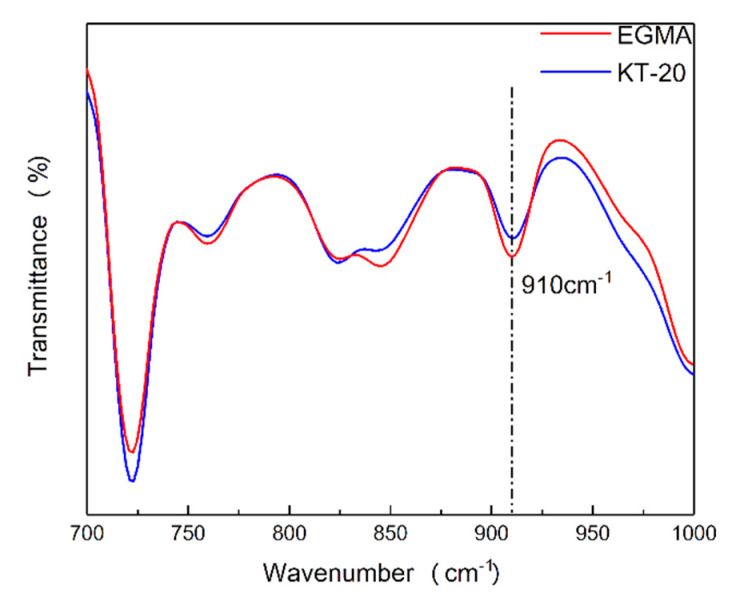
Characteristic absorption peak at 910 cm^−1^.

**Figure 8 materials-13-02094-f008:**
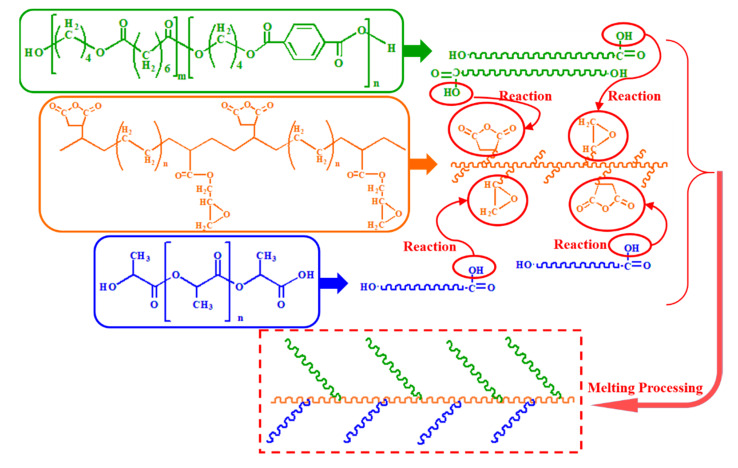
Chemical reaction in blend preparation process (taking the compatibilizer KT-20 as an example).

**Figure 9 materials-13-02094-f009:**
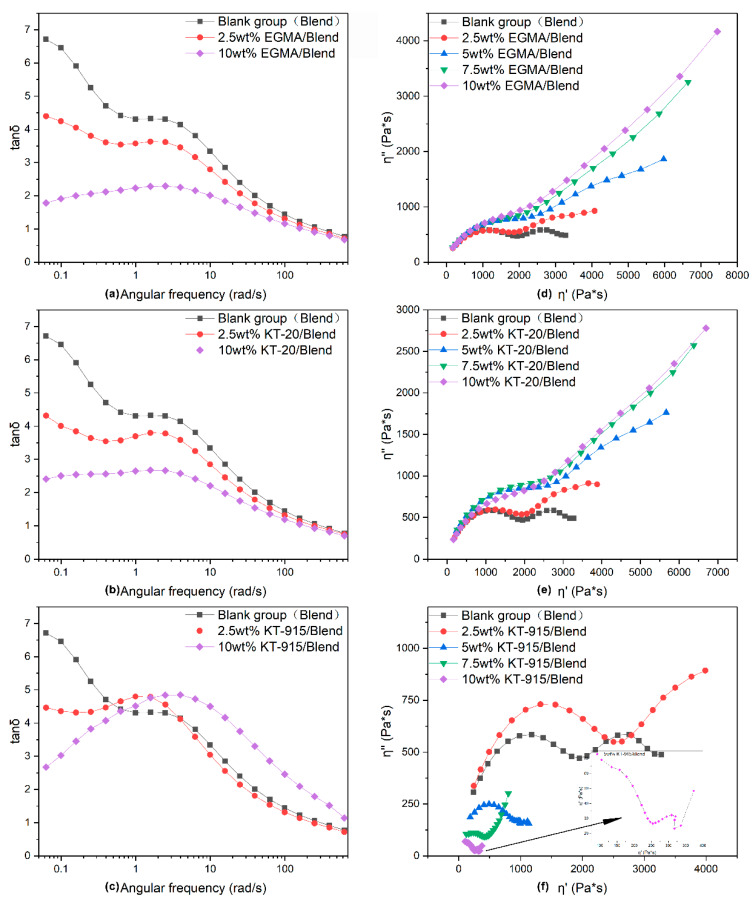
The loss factor-angular frequency curves (left column) and dynamic viscosity—loss viscosity curves (right column).

**Figure 10 materials-13-02094-f010:**
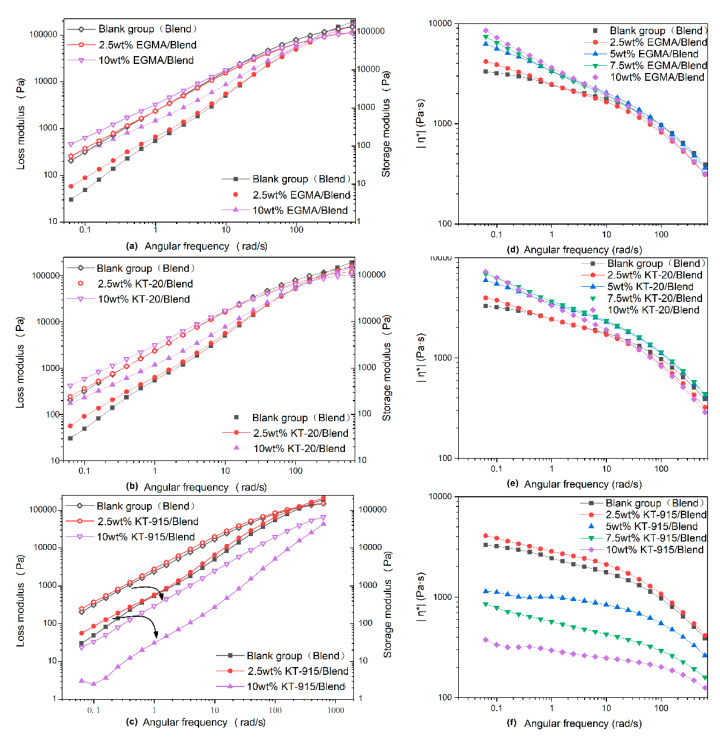
Loss modulus and storage modulus-angular frequency curve (left column), and complex viscosity-angular frequency curve (right column).

**Figure 11 materials-13-02094-f011:**
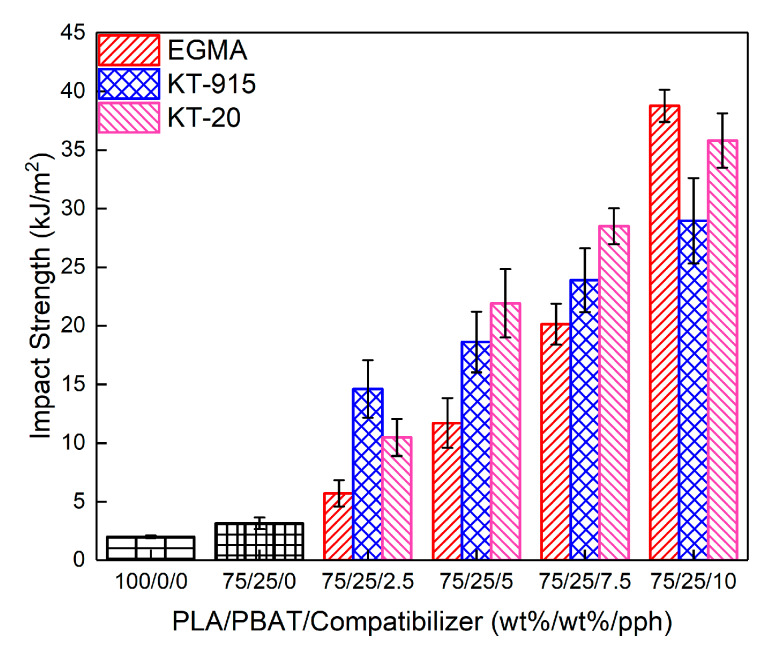
Impact strength of all components.

**Table 1 materials-13-02094-t001:** Raw material information and basic properties.

Material	Trademark	Density(g/cm^−3^)	MFI(190 °C, 2.16 kg)	Factory
PLA (semi-crystalline material containing 2% D-lactide units)	SC M310–A2-HP	1.24	6 g/10 min	COFCO Biomaterial Co., Ltd. Jilin China
PBAT	PBAT	1.20	43 g/10 min	Kingfa SCI. & TECH. Co., Ltd. Guangzhou, China
Compatibilizer1: EGMA	AX 8900	0.95	6.0 g/10 min	Arkema Investment Co., Ltd. (Shanghai, China)
Compatibilizer2: KT-20	KT-20	0.94	1.3 g/10 min	Shenyang Ketong Plastic Co., Ltd. Shenyang, China
Compatibilizer3: KT-915	KT-915	0.85	0.87 g/10 min	Shenyang Ketong Plastic Co., Ltd. Shenyang, China

(Note: Mn of poly(lactic acid) (PLA), poly(butylene adipate-co-terephthalate) (PBAT), KT-915, KT-20 and EGMA were provided by factories as follows: 2.50 ∗ 10^5^ g/mol, 2.12 ∗ 10^5^ g/mol, 1.08 ∗ 10^4^ g/mol, 5.42 ∗ 10^4^ g/mol and 3.25 ∗ 10^4^ g/mol.).

**Table 2 materials-13-02094-t002:** Component mass ratio.

Abbreviation	Various Matrix Weight
Blank group (Blend)	PLA/PBAT (75 wt%/25 wt%)
2.5 wt% EGMA/Blend	PLA/PBAT/EGMA (75 wt%/25 wt%/2.5 wt%)
5 wt% EGMA/Blend	PLA/PBAT/EGMA (75 wt%/25 wt%/5 wt%)
7.5 wt% EGMA/Blend	PLA/PBAT/EGMA (75 wt%/25 wt%/7.5 wt%)
10 wt% EGMA/Blend	PLA/PBAT/EGMA (75 wt%/25 wt%/10 wt%)
2.5 wt% KT-915/Blend	PLA/PBAT/KT-915 (75 wt%/25 wt%/2.5 wt%)
5 wt% KT-915/Blend	PLA/PBAT/KT-915 (75 wt%/25 wt%/5 wt%)
7.5 wt% KT-915/Blend	PLA/PBAT/KT-915 (75 wt%/25 wt%/7.5 wt%)
10 wt% KT-915/Blend	PLA/PBAT/KT-915 (75 wt%/25 wt%/10 wt%)
2.5 wt% KT-20/Blend	PLA/PBAT/KT-20 (75 wt%/25 wt%/2.5 wt%)
5 wt% KT-20/Blend	PLA/PBAT/KT-20 (75 wt%/25 wt%/5 wt%)
7.5 wt% KT-20/Blend	PLA/PBAT/KT-20 (75 wt%/25 wt%/7.5 wt%)
10 wt% KT-20/Blend	PLA/PBAT/KT-20 (75 wt%/25 wt%/10 wt%)

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
