# Peer review of "Effect of Different Compatibilizers on the Properties of Poly (Lactic Acid)/Poly (Butylene Adipate-Co-Terephthalate) Blends Prepared under Intense Shear Flow Field"

_materials, 2020, doi:10.3390/ma13092094_

Round 1

Reviewer 1 Report

The manuscript presents the idea of using compatibilizer to make polymer blend of 75% PLA and 25% PBAT. The manuscript present interesting aspects of dynamic rheological properties of polymer blends. This manuscript should be accepted only after minor revision. Here are my comments.

Reactive extrusion of polymer components to make a compatible blend is interesting, however, there is less clarity about the process used in this study such as, details of the screw design, configuration, length to diameter ratio, how much residence time at the given screw speed, etc. It’s also interesting if authors can provide some data or make a comment on how changing shear force in reactive extrusion can make any difference on blend compatibility. Typically, it does. For compatibilization purpose, 10% compatibilizer is too much, however, it interesting to see how the impact strength keep increasing with the amount of compatibilizer in all cases. What happened if you increase the compatibilizer more than 10%? The storage modulus, loss modulus, and complex viscosity indicate that only epoxied groups are reacting during reactive extrusion and anhydride groups remain unreactive. Could you please include storage modulus vs temperature also? It would be interesting to see any reinforcement effect at higher temperature, especially in case of KT20. Can you comment what temperature you can go any further in reactive extrusion to react the compatibilizer with polymer components. What are the molecular weights of compatibilizers and what role molecular weight plays in blend properties like impact strength and dynamic rheological performance? In Figure 5, 10wt% KT-20/blend shows maximum decrease in hydroxyl functionalities; however, the impact strength is lower than EGMA. Is it because of the characteristics of compatibilizer molecules?

Reviewer 2 Report

Reviewed manuscript is interesting for biodegradable polyester blends fabrication and should be published. However, before accepting several problems have to be included.

Present/explain, please, in the aim of the work what is new in this manuscript comparing with the ref. 17 – 24. (Sorry I have problems with accessing some of these papers).
You wrote about the beneficial effect of the compatibilizer containing epoxy and maleic anhydride reactive groups, i.e., EGMA and PLA-g-MA but in present manuscript you also used these oligomers (or polymers – there is no data concerning molar mass of raw materials). Moreover, do you think that the size of the machine will affect the quality of the blend? Please explain these problems in revised manuscript. You proposed in Fig. 7 chemical reactions occurring in blend preparation process. They seems to be correct, however, it should be proved using other techniques than FTIR, e.g., 1HNMR or at least SEC experiment. In SEC of blend obtained with compatibilizer you should observe different result than received for produced without EGMA or PLA-g-MA. I hope you will can prove in this experiment the molar mass increase. NMR basing on the structure of the resulting products will prove the type of reactions which occurs in your blending. Are you sure that the values presented in Table 1, third column, are specific gravity. They seems to look like density (see units gcm3). Careful correction of the English language is required. I advise you ask to do it native speaker, material chemist.

Reviewer 3 Report

The manuscript "Effect of Different Compatibilizers on the Properties of Poly(lactic Acid)/Poly(butylene adipate-co-terephthalate) Blends Prepared under Intense Shear Flow Field" is presenting an interesting scientific and practical topic concerning the improvement of mechanical properties of the bio-based PLA via polymer blending

The toughening modification of PLA by blending with flexible polymers or plasticizers has been extensively investigated.

The present work focusses on an important aspect concerning the immiscible polymler blends, namely the influence of different compatibilizers on the PLA/PBAT blends generated by intensive shear flow.

The experimental data are interesting and deserved to be published, after a major revision of the manuscript.

The introduction contains several imprecisions, starting with the first sentence "Poly(lactic acid) (PLA), a green polymer material, can be fermented by glucose and further polymerized"  - mixing confusing information about the monomer and polymer.

Another example is the sentence lines 45-46.

Reference 21-23 are not significant for the present topic. For instance, [23] concerns a polylactic acid racemic blends, a very special case of PLA, and the mechanisms involved in such type of blends are very specific and not applicable here.

In addition, in the present work, the type PLA is not explicitely indicated (PLLA, PDLA - with %L ?), or the final mechanical properties are also depending on the type of virgin PLA used in this study.

The experimental part should be re-organized, since the morphology obtained by intense shear flow, in presence of different compatibilizers, is the one explaining the final blends characteristics, including the impact strengh.

Information about the different morphologies obtained for a given high shear flow profile along the extruder (and same T°C processing), and the type of interactions at the PLA/PBAT interfaces are given directly by SEM and indirectly by FT-IR. Presenting these experimental results first will allow a better discussion of the rheological measurement and will offer improved explainations concerning the effect of the chemical structure and percentage of compatibilizers on the mechanical properties.

Furthermore, it is also important to indicate the exact procedure to add the three different compatibilizers inside the polymer blend, and to justify the choice of premixtures, if any.

The SEM images for compatibilized polymer blends should be presented at higher magnifications and organized by type of compatibilizers, to allow effective analysis of the average size of PBAT inclusions, dispersion distribution of the size of inclusions and the interface characteristics - depending on the different compatibilizers under study.

In Figure 3 - data for virgin PBAT are missing, and the text concerning this figure is rather descriptive than explicative. Same observation is also applicable to the discussion on rheological experimental data.

The English language should be also systematically improved.

In consequence, I recommand a major revision of the present manuscript.

Round 2

Reviewer 2 Report

I think the manuscript may be published. However the English should be additionally polished e.g. see first sentence of Introduction: Poly(lactic acid) (PLA), a green polymer material can be polymerized by lactic acid which fermented by glucose. What Authors were thinking about when writting this sentence? PLA can be synthesized via polymerization of LA or lactide. PLA cannot be polymerized.
Please ask also polymer or organic chemist for reading the manuscript.

Author Response

Thanks for your kindly comment very much.

Point:I think the manuscript may be published. However the English should be additionally polished e.g. see first sentence of Introduction: Poly(lactic acid) (PLA), a green polymer material can be polymerized by lactic acid which fermented by glucose. What Authors were thinking about when writting this sentence? PLA can be synthesized via polymerization of LA or lactide. PLA cannot be polymerized. 
Please ask also polymer or organic chemist for reading the manuscript.

Responds: The errors in the manuscript have been corrected, and the language has been further revised under the guidance of polymer chemists.

Reviewer 3 Report

The revised version of the manuscript is largely improved.

The only minor change I would recommend is the use of %wt rather than pph - however, as long as the same concentration unit is used all over the manuscript, it is OK.

I recommend this improved manuscript-v2 version for publication.

Author Response

Thanks for your kindly comment.

Point:

The revised version of the manuscript is largely improved.

The only minor change I would recommend is the use of %wt rather than pph - however, as long as the same concentration unit is used all over the manuscript, it is OK.

I recommend this improved manuscript-v2 version for publication.

Respond:

All of the "pph" have been corrected to "wt%" in the revised manuscript.